# The Structural Rule Distinguishing a Superfold: A Case Study of Ferredoxin Fold and the Reverse Ferredoxin Fold

**DOI:** 10.3390/molecules27113547

**Published:** 2022-05-31

**Authors:** Takumi Nishina, Megumi Nakajima, Masaki Sasai, George Chikenji

**Affiliations:** 1Department of Applied Physics, Nagoya University, Nagoya 464-8601, Japan; nishina@tbp.ap.pse.nagoya-u.ac.jp (T.N.); nakajima@tbp.cse.nagoya-u.ac.jp (M.N.); 2Department of Complex Systems Science, Nagoya University, Nagoya 464-8601, Japan; 3Fukui Institute for Fundamental Chemistry, Kyoto University, Kyoto 606-8501, Japan

**Keywords:** protein design, reverse fold, minimum frustration

## Abstract

Superfolds are folds commonly observed among evolutionarily unrelated multiple superfamilies of proteins. Since discovering superfolds almost two decades ago, structural rules distinguishing superfolds from the other ordinary folds have been explored but remained elusive. Here, we analyzed a typical superfold, the ferredoxin fold, and the fold which reverses the N to C terminus direction from the ferredoxin fold as a case study to find the rule to distinguish superfolds from the other folds. Though all the known structural characteristics for superfolds apply to both the ferredoxin fold and the reverse ferredoxin fold, the reverse fold has been found only in a single superfamily. The database analyses in the present study revealed the structural preferences of αβ- and βα-units; the preferences separate two α-helices in the ferredoxin fold, preventing their collision and stabilizing the fold. In contrast, in the reverse ferredoxin fold, the preferences bring two helices near each other, inducing structural conflict. The Rosetta folding simulations suggested that the ferredoxin fold is physically much more realizable than the reverse ferredoxin fold. Therefore, we propose that minimal structural conflict or minimal frustration among secondary structures is the rule to distinguish a superfold from ordinary folds. Intriguingly, the database analyses revealed that a most stringent structural rule in proteins, the right-handedness of the βαβ-unit, is broken in a set of structures to prevent the frustration, suggesting the proposed rule of minimum frustration among secondary structural units is comparably strong as the right-handedness rule of the βαβ-unit.

## 1. Introduction

A principal goal of protein science is to elucidate the relationship among sequences, structures, and functions [1,2]. Toward such a goal, remarkable progress has been achieved in structure prediction from the knowledge of amino-acid sequences [3,4]. Also, in protein design, which is a reverse problem of structure prediction, elucidation of design principles [5,6,7] led to an increasing number of successful examples to find amino-acid sequences that can fold into the designed structures [5,6,8,9,10,11,12]. Here, for further advancing the design technology, it is crucial to develop a systematic method to distinguish less designable structures and highly designable ones into each of which a large number of different sequences can fold [13]. Investigating the occurrence of structural folds among natural proteins provides a clue to this problem [14,15,16,17,18]. An ordinary fold appears in only one or a few superfamilies, but a particular fold is shared by a large number of superfamilies; such a particular fold was called a superfold [19]. Here, a superfamily is defined as the largest group of proteins for which common ancestry can be inferred [20]. Superfolds are rare in the entire fold categories but are robust against mutations, suggesting superfolds represent highly designable structures. Each superfold corresponds to many different functions, in sharp contrast to the ordinary folds showing the nearly one-to-one correspondence between fold and function.

Since the discovery of superfolds [19], features distinguishing superfolds from the other ordinary folds have been explored, leading to the several empirical rules that characterize the superfolds, some of which are (1) frequent appearance of super secondary structures [21], (2) avoidance of mixing parallel and anti-parallel β-sheets [14], (3) infrequent jumps between β-strands [16], and (4) high structural symmetry [22]. However, examples of ordinary folds satisfy the rules from (1) through (4), showing the need for further rules to distinguish superfolds. The reverse ferredoxin fold is such an example. The ferredoxin fold, a typical superfold, comprises four β-strands connected in the order and directions as designated in Figure 1A. The reverse ferredoxin fold reverses the N to C terminus direction from the ferredoxin fold (Figure 1B). According to the SCOPe classification [23,24], the ferredoxin fold is found in 62 superfamilies, whereas the reverse ferredoxin fold is found only in one superfamily. Therefore, the reverse ferredoxin fold is not a superfold, but both the ferredoxin fold and the reverse ferredoxin fold satisfy the rules (1) through (4). Other examples show the significant difference between the fold and the reverse fold in the number of occurrences in the spectrum of families [15]. The reason for this difference between folds and reverse folds remains elusive; there have been arguments suggesting physical or functional necessities to avoid the reverse folds [15] and those suggesting the bias occasionally acquired in evolutionary history [25].

Here, we explored the factor to distinguish superfolds from the ordinary folds by comparing the ferredoxin fold and the reverse ferredoxin fold as a case study. By analyzing the database, we found the structural tendency shown by the αβ-unit and βα-unit, suggesting that the structure comprises multiple αβ- and βα-units should satisfy a rule to minimize the conflict between structural tendencies of these units. We show that the ferredoxin fold satisfies this rule for minimal conflict or frustration, whereas the reverse ferredoxin fold does not. We also performed the Rosetta folding simulations to test the foldability of structures [5]; the test results suggested that the ferredoxin fold is physically much more realizable than the reverse ferredoxin fold. Thus, we propose that the minimum frustration rule to consistently satisfy the structural preference of multiple parts of the protein is a rule to distinguish superfolds from ordinary folds.

## 2. Results

### 2.1. Occurrence Frequency of Topologies

Previous analyses showed that the ferredoxin fold is frequently found, whereas the reverse ferredoxin fold is rare among protein families [17,25]. We confirmed this imbalance in the most recent version of a semi-manually curated database, ECOD (version 20210511: develop280), which hierarchically classifies protein domains according to homology, reflecting their evolutionary relationship [26]. ECOD has been frequently updated, suited to estimating the most recent number of homology groups having a topology on which we focus. The ECOD database classifies homologous protein domains according to categories of family and homology. The family (F) group consists of evolutionarily related protein domains with substantial sequence similarity, and the homology (H) group comprises multiple F-groups having functional and structural similarities. The H-group corresponds to the superfamily in the other structural databases, SCOP [27] and CATH [28]. The X-group in ECOD comprises multiple H-groups that share similar features in the structure but lack a convincing evidence for homology. In this study, we used the 99% sequence identity representatives in ECOD as the dataset for the analyses.

We detected secondary structures and hydrogen bonds in protein domains recorded in the dataset using STRIDE [29]. Then, based on the thus found hydrogen-bond pattern among β-strands, we defined the β-sheet topology as in Ref. [15]; we describe the β-sheet topology by representing the strand directions with up and down arrows with the sequential number from the N- to C-termini (4132, for example). Then, topology *T* of the ferredoxin fold is $T=4_{↓}1_{↑}3_{↓}2_{↑}$ (Figure 1A) and topology *T* of the reverse ferredoxin fold is $T=1_{↑}4_{↓}2_{↑}3_{↓}$ (Figure 1B).

We estimated the occurrence frequency OF(T) of a given topology *T* by summing the occupation ratio OR(T,i) of protein domains having *T* in the *i*th H-group as
(1)$$\begin{array}{c} OF\left(T\right)=\sum_{i=1}^{N_{\mathrm{homology}}}OR\left(T,i\right), \end{array}$$
where $N_{\mathrm{homology}}$ is the total number of H groups in the dataset, and
(2)$$\begin{array}{c} OR\left(T,i\right)=\frac{1}{N_{\mathrm{family}}\left(i\right)}\sum_{j=1}^{N_{\mathrm{family}}\left(i\right)}\frac{N_{\mathrm{domain}}\left(T,i,j\right)}{N_{\mathrm{domain}}\left(i,j\right)}. \end{array}$$

Here, $N_{\mathrm{domain}}\left(T,i,j\right)$ is the number of protein domains having topology *T* in the *j*th F-group, which belongs to the *i*th H-group in the dataset. $N_{\mathrm{domain}}\left(i,j\right)=\sum_{T}N_{\mathrm{domain}}\left(T,j\right)$ is the total number of protein domains in the *j*th F-group, and $N_{\mathrm{family}}\left(i\right)$ is the number of F-groups in the *i*th H-group. Figure 1C shows that the occurrence frequency of the ferredoxin topology, $OF(4_{↓}1_{↑}3_{↓}2_{↑})$, is more than 10 times larger than the occurrence frequency of the reverse ferredoxin topology, $OF(1_{↑}4_{↓}2_{↑}3_{↓})$, confirming the previously reported ubiquity of the ferredoxin fold and the rareness of the reverse ferredoxin fold [17,25].

Here, we should note that topology has often been classified with ECOD in terms of X-groups; for example, an X-group called “alpha-beta plaits” has been regarded as the group representing the ferredoxin topology. However, we used STRIDE for a more precise topological classification instead of the X-group classification. Therefore, the OF(T) defined in Equation (Equation 1) does not precisely correlate with the number of H-groups in the X-group. Tetracycline resistance protein, tetM (PDB ID: 3J25), for example, belongs to the X-group of alpha-beta plaits, but we did not count tetM as a ferredoxin-topology protein because STRIDE identifies only two β-strands in tetM. Similarly, surface-layer (S-layer) protein (PDB ID: 3CVZ) belongs to the reverse ferredoxin X-group in ECOD, but we did not count S-layer protein as a protein with the reverse-ferredoxin fold because STRIDE identifies a topology $1_{↑}5_{↑}4_{↓}2_{↑}3_{↓}$ for S-layer protein instead of $1_{↑}4_{↓}2_{↑}3_{↓}$. See Appendix A for the structure of tetM and S-layer protein.

We examine the minimal structural units that induce the difference between $4_{↓}1_{↑}3_{↓}2_{↑}$ and $1_{↑}4_{↓}2_{↑}3_{↓}$. We consider the topology in which the C-terminal strand (β-strand 4) is deleted from the ferredoxin topology by retaining the α-helix connecting β-strands 4 and 3 in the structure, and write the thus obtained topology as $1_{↑}3_{↓}2_{↑}+C-\mathrm{term}\;\alpha$. We also consider the topology in which the C-termα is further deleted from $1_{↑}3_{↓}2_{↑}+C-\mathrm{term}\;\alpha$ and write such a topology as $1_{↑}3_{↓}2_{↑}$. Similarly, we consider the topology in which the N-terminal strand (β-strand 1) is deleted from the reverse ferredoxin topology by retaining the α-helix connecting β-strands 1 and 2 in the structure. Then, we renumber the strands as 4,2,3→3,1,2, and write the thus-obtained topology as $3_{↓}1_{↑}2_{↓}+N-\mathrm{term}\;\alpha$, which is the reverse of $1_{↑}3_{↓}2_{↑}+C-\mathrm{term}\;\alpha$. We also consider the topology in which the N-termα is further deleted from $3_{↓}1_{↑}2_{↓}+N-\mathrm{term}\;\alpha$ and write such a topology as $3_{↓}1_{↑}2_{↓}$, which is the reverse of $1_{↑}3_{↓}2_{↑}$.

We consider protein domains whose entire (not the partial) structure has the topology $1_{↑}3_{↓}2_{↑}+C-\mathrm{term}\;\alpha$ or $3_{↓}1_{↑}2_{↓}+N-\mathrm{term}\;\alpha$, and calculated occurrence frequencies, $OF(1_{↑}3_{↓}2_{↑}+C-\mathrm{term}\;\alpha )$ and $OF(3_{↓}1_{↑}2_{↓}+N-\mathrm{term}\;\alpha )$ (Figure 1D). We should note that with the topology of $1_{↑}3_{↓}2_{↑}+C-\mathrm{term}\;\alpha$, the C-termα can lie on either side of the β-sheet plane. However, in the ferredoxin fold, this helix is always on the same side of the plane as the α-helix of the βαβ-unit consisting of β-strands 1 and 2; therefore, we here calculated $OF(1_{↑}3_{↓}2_{↑}+C-\mathrm{term}\;\alpha )$ for the structures in which the C-termα is on the same side of the plane as the α-helix of the βαβ-unit. Similarly, we calculated $OF(3_{↓}1_{↑}2_{↓}+N-\mathrm{term}\;\alpha )$ for structures in which the N-termα is on the same side of the β-sheet plane as the α-helix of the βαβ-unit consisting of β-strands 2 and 3. See the Materials and Methods section for the way to judge which side of the plane the terminal helix lies in a given structure in calculating OFs. Figure 1D shows that $OF(1_{↑}3_{↓}2_{↑}+C-\mathrm{term}\;\alpha )$ is significantly larger than $OF(3_{↓}1_{↑}2_{↓}+N-\mathrm{term}\;\alpha )$, suggesting that the determining structural factor distinguishing the ferredoxin fold and the reverse ferredoxin fold exists in the difference between $1_{↑}3_{↓}2_{↑}+C-\mathrm{term}\;\alpha$ and $3_{↓}1_{↑}2_{↓}+N-\mathrm{term}\;\alpha$. The population of the structures with two helices lying on the opposite side of the β-sheet plane is small in the $1_{↑}3_{↓}2_{↑}+C-\mathrm{term}\;\alpha$ topology and in the $3_{↓}1_{↑}2_{↓}+N-\mathrm{term}\;\alpha$ topology, and there is no significant difference between occurrence frequencies of two topologies for those structures with helices lying on the opposite side of the plane. The large difference between two topologies only appear for structures in which two helices lie on the same side of the plane (Appendix A).

Similarly, we calculated occurrence frequencies, $OF(1_{↑}3_{↓}2_{↑})$ and $OF(3_{↓}1_{↑}2_{↓})$ (Figure 1E), showing that $OF(3_{↓}1_{↑}2_{↓})$ is mildly larger than $OF(1_{↑}3_{↓}2_{↑})$. These results suggest that the determinant structural factor that induces the difference between $4_{↓}1_{↑}3_{↓}2_{↑}$ and $1_{↑}4_{↓}2_{↑}3_{↓}$ is in the difference between $1_{↑}3_{↓}2_{↑}+C-\mathrm{term}\;\alpha$ and $3_{↓}1_{↑}2_{↓}+N-\mathrm{term}\;\alpha$. Addition of the C-termα-helix to $1_{↑}3_{↓}2_{↑}$ and addition of the N-termα-helix to $3_{↓}1_{↑}2_{↓}$ bring about the difference in the occurrence frequency between the ferredoxin topology and the reverse ferredoxin topology. Hereafter, the ferreoxin fold and the $1_{↑}3_{↓}2_{↑}+C-\mathrm{term}\;\alpha$ topology are referred to collectively as the ferredoxin-type topology, and the reverse ferredoxin fold and the $3_{↓}1_{↑}2_{↓}+N-\mathrm{term}\;\alpha$ topology are referred to collectively as the reverse ferredoxin-type topology.

### 2.2. Conflict between Structural Preferences of αβ- and βα-Units

Because positions of the αβ- and βα-units are different in $1_{↑}3_{↓}2_{↑}+C-\mathrm{term}\;\alpha$ and $3_{↓}1_{↑}2_{↓}+N-\mathrm{term}\;\alpha$ (Figure 1A,B), analyses on these structural units should give critical insights on the difference between $1_{↑}3_{↓}2_{↑}+C-\mathrm{term}\;\alpha$ and $3_{↓}1_{↑}2_{↓}+N-\mathrm{term}\;\alpha$. For the structural analyses of these units, we defined the distance *x* between the plane of the β-pleats in the strand and the α-helix (Figure 2A). See the Materials and Methods section for the precise definition of *x*. We derived the distribution of *x* by analyzing the dataset culled from PDB with constraints of the sequence identity <30%, the finer resolution than 2.0 Å, and the *R*-factor <0.25 [30]. For the statistical analyses, we selected typical αβ- and βα-units following the criterion of Ref. [31]; we used the structural units satisfying the conditions that the linker loop between α-helix and β-strand is shorter than five-residue length and the angle between α-helix and β-strand is less than 60 ${}^{\circ }$.

Figure 2B shows the distribution of *x* obtained by the dataset analyses. The distribution of *x* in the βα-unit peaked at 2∼4 Å, whereas the distribution of *x* in the αβ-unit peaked at ∼0 Å, showing a distinct tendency of positive *x* in the βα-unit. This positive *x* distribution implies the tendency of shifting the α-helix toward the direction of blue arrows in Figure 2C. In the $1_{↑}3_{↓}2_{↑}+C-\mathrm{term}\;\alpha$ structure, this shift separates the C-termα-helix from the helix in the βαβ structure, while in the $3_{↓}1_{↑}2_{↓}+N-\mathrm{term}\;\alpha$ structure, the shift induces collision of the N-termα-helix against the helix in the βαβ structure when two helices are on the same side of the β-sheet surface. Therefore, the structural conflict arising between two helices destabilizes the $3_{↓}1_{↑}2_{↓}+N-\mathrm{term}\;\alpha$ structure; and hence, destabilizes the reverse ferredoxin fold.

We can quantitatively assess how the difference in the distribution of the distance *x* in Figure 2B determines the absence/presence of the structural conflict. We write *x* in the βα-unit and the αβ-unit as $x_{\beta \alpha }$ and $x_{\alpha \beta }$, respectively. Considering that a typical distance between two adjacent β-strands in a β-sheet is 4.5 Å [32], the distance between two helices in the ferredoxin-type topology is $x_{\beta \alpha }-x_{\alpha \beta }+4.5$ Å. Similarly, the distance between two helices in the reverse ferredoxin-type topology is $x_{\alpha \beta }-x_{\beta \alpha }+4.5$ Å (Figure 2D). Because the helix diameter is approximately 11.0 Å [33], the necessary condition to avoid the collision of two helices is $x_{\beta \alpha }-x_{\alpha \beta }+4.5\;Å>11$Å for the ferredoxin-type topology and $x_{\alpha \beta }-x_{\beta \alpha }+4.5\;Å>11$Å for the reverse ferredoxin-type topology. In Figure 2E, the region satisfying three conditions at the same time is designated by a green triangle on a two-dimensional plane of $x_{\beta \alpha }$ and $x_{\alpha \beta }$: (i) the necessary condition to avoid the collision, (ii) the condition of frequency >5% in the frequency distribution of $x_{\beta \alpha }$ in Figure 2B, and (iii) the condition of frequency >5% in the frequency distribution of $x_{\alpha \beta }$ in Figure 2B. The thus-defined green triangle, i.e., the realizable area to avoid the collision, is extremely narrow in the reverse ferredoxin-type topology, whereas it is wide in the ferredoxin-type topology. Figure 2E shows that the occurrence frequency of $\left(x_{\beta \alpha },x_{\alpha \beta }\right)$ in the ECOD database is large around the green triangle in the ferredoxin-type fold, while the frequency is small everywhere on the plane of $\left(x_{\beta \alpha },x_{\alpha \beta }\right)$ in the reverse ferredoxin-type fold. Thus, the shift of 2∼4 Å in distributions in Figure 2B is a determining factor for the realizability of the structure. In the reverse ferredoxin-type topology, the structures are realized by breaking at least one of three conditions (i)–(iii). Different ways of breaking the conditions in the reverse ferredoxin-type topology make the distribution scattered on the $\left(x_{\beta \alpha },x_{\alpha \beta }\right)$ plane in Figure 2E. Appendix A shows example proteins with the reverse ferredoxin topology showing uncommon configuration of the βα- or αβ-unit.

We should note that the results shown in Figure 2B,E are the plots for proteins with loops shorter than five-residue length. The longer loops allow the structural variety to obscure the realizability conditions in Figure 2B,E. However, the stability of native structures inversely correlates to the loop length [34,35], making the proteins having the longer loops rare. See Appendix A for the distribution of the loop length found in the ECOD database. Here, it is sufficient to consider non-rare proteins with short enough loops for clarifying how the ferredoxin-type topology is much more realizable than the reverse ferredoxin-type topology.

### 2.3. Minimum Frustration Rule

The dataset analyses showed that the structural preference of αβ- and βα-units leads to the structural conflict in the $3_{↓}1_{↑}2_{↓}+N-\mathrm{term}\;\alpha$ structure, while the conflict is avoided in the $1_{↑}3_{↓}2_{↑}+C-\mathrm{term}\;\alpha$ structure. We examined the effect of presence/absence of the structural conflict by performing the Rosetta folding simulations. In these simulations, we substituted all the residues in the model to Valine, and assembled the fragments of one-, three-, or nine-residue length, which have the compatible main-chain dihedral angles with the secondary structures in the blueprints designated in Figure 3. We used the all-Valine sequence to focus on the role of structural consistency among the assembled fragments instead of the effects of the residue-specific interactions. We regard structures generated through the simulations as compatible structures when they have low energy and the same topology as the blueprint. For each blueprint, we performed the fragment-assembly simulation 10,000 times and counted how many compatible structures were obtained through simulations. Koga et al. showed that the topology designated by the blueprint is physically realizable by avoiding the structural conflict when the number of the obtained compatible structures is large, while it is physically unrealizable with the structural inconsistency when the number is small [5]. See the Materials and Methods section for the details of the simulations.

Figure 3A shows the number of structures compatible with the $3_{↓}1_{↑}2_{↓}+N-\mathrm{term}\;\alpha$ topology and the number of structures compatible with the $1_{↑}3_{↓}2_{↑}+C-\mathrm{term}\;\alpha$ topology. The compatible structures were 229 and 10 for the $1_{↑}3_{↓}2_{↑}+C-\mathrm{term}\;\alpha$ topology and the $3_{↓}1_{↑}2_{↓}+N-\mathrm{term}\;\alpha$ topology, respectively, showing the $1_{↑}3_{↓}2_{↑}+C-\mathrm{term}\;\alpha$ topology is much more realizable than the $3_{↓}1_{↑}2_{↓}+N-\mathrm{term}\;\alpha$ topology. We performed the same test for the $1_{↑}3_{↓}2_{↑}$ topology and the $3_{↓}1_{↑}2_{↓}$ topology. Figure 3B shows that the number of compatible structures for the $1_{↑}3_{↓}2_{↑}$ topology is almost same as the number of compatible structures for the $3_{↓}1_{↑}2_{↓}$ topology, indicating that there is no significant difference between the realizability of these topologies. Figure 3A,B are qualitatively same as Figure 1D,E, showing that the difference in the realizability of the $1_{↑}3_{↓}2_{↑}+C-\mathrm{term}\;\alpha$ topology and the $3_{↓}1_{↑}2_{↓}+N-\mathrm{term}\;\alpha$ topology arises from absence/presence of the conflict between local structural units.

Combined analyses of databases and Rosetta folding simulations showed that the structural conflict or frustration is minimized in the largely realizable topology, which characterizes the superfold; therefore, we propose that the minimum frustration among local preferences of secondary structures is the rule to distinguish a superfold from the ordinary folds.

## 3. Discussion

In this study, we proposed a rule that the minimum frustration among local structural preferences of secondary structures is the necessary condition for superfolds. In this section, we discuss the meaning of this rule by explaining how the rule predicts occurrence frequency of other structures, the relation of the rule with the other design rule, and the relation with protein function.

### 3.1. Occurrence Frequency of Other Structures

The present analyses of the ferredoxin fold and the reverse ferredoxin fold showed that the frequently occurring topology is designed to minimize frustration among multiple secondary-structure units that lie near each other on the same side the β-sheet plane. We can examine whether this rule predicts the occurrence frequency of other structures in the dataset. Figure 4A–D are four examples of pairs of topologies; in each pair, one is the topology minimizing frustration, and the other is its reverse topology exhibiting frustration. We should note that pairs in Figure 4B–D have the same arrangement of β-strands but have different connections of terminal α-helices showing different topologies. Our rule of minimum frustration predicts that the topology shown on the left side in each pair in Figure 4 is more realizable than the topology on the right side. We counted the occurrence frequency of these topologies in the dataset and found a significant difference as expected. In particular, we found the zero occurrence frequency of the frustrated topology in Figure 4D. The absence of this topology is reasonable because the frustrated topology of Figure 4D has two positions of structural collisions between helices, whereas the other frustrated topologies in Figure 4A–C show only a single collision in each. These results support our proposal that the minimum frustration among secondary structures is the requirement for the frequently occurring topologies; therefore, the necessary condition for the superfolds.

### 3.2. The Left-Handed βαβ-Unit Is Selectively Found in the $3_{↓}1_{↑}2_{↓}+N-\mathrm{term}\alpha$ Structures

We showed that the collision between two helices arising from the structural preference of nearby αβ- and βα-units decreases the occurrence frequency of the $3_{↓}1_{↑}2_{↓}+N-\mathrm{term}\;\alpha$ topology. However, this collision disappears when the two helices lie on the opposite side of the β-sheet surface. Such configurations are possible in two different ways. One is the configuration that the βαβ-unit consisting of β-strands 2 and 3 is right-handed and the terminal helix is on the opposite side; we have a small number of such examples in the dataset as shown in Appendix A. The other is the configuration that the βαβ-unit is left-handed with the terminal helix in the position similar to that in the reverse ferredoxin fold. Here, we cannot expect the frequent occurrence of the latter structure because more than 98% of the known βαβ-unit structures are right-handed [14,36,37,38]. Indeed, in our dataset derived from ECOD, there is no left-handed βαβ-unit in protein domains with the $1_{↑}3_{↓}2_{↑}+C-\mathrm{term}\;\alpha$ or the $3_{↓}1_{↑}2_{↓}+N-\mathrm{term}\;\alpha$ topology.

However, in the dataset, we found a small number of left-handed βαβ-units in protein domains having the extended structures including $1_{↑}3_{↓}2_{↑}+C-\mathrm{term}\;\alpha$ or $3_{↓}1_{↑}2_{↓}+N-\mathrm{term}\;\alpha$ as a partial structure (Figure 5B,C). See the Materials and Methods section for the method to detect the left-handed βαβ-unit in the dataset. Figure 5A shows occurrence frequencies of domains in the dataset having more than four β-strands and include the $1_{↑}3_{↓}2_{↑}+C-\mathrm{term}\;\alpha$ or the $3_{↓}1_{↑}2_{↓}+N-\mathrm{term}\;\alpha$ topology as their partial structure. For these extended domains, we counted occurrence frequencies separately for those having a left-handed βαβ-unit, $OF(\mathrm{Extended}-1_{↑}3_{↓}2_{↑}+C-\mathrm{term}\;\alpha ;\mathrm{Left})$ and $OF(\mathrm{Extended}-3_{↓}1_{↑}2_{↓}+N-\mathrm{term}\;\alpha ;\mathrm{Left})$, and for those having the right-handed βαβ-unit, $OF(\mathrm{Extended}-1_{↑}3_{↓}2_{↑}+C-\mathrm{term}\;\alpha ;\mathrm{Right})$ and $OF(\mathrm{Extended}-3_{↓}1_{↑}2_{↓}+N-\mathrm{term}\;\alpha ;\mathrm{Right})$. We found OF(Extended-$1_{↑}3_{↓}2_{↑}$+C-term α; Right) = 73.8, OF(Extended-$1_{↑}3_{↓}2_{↑}$+C-term α; Left) = 0.5, OF(Extended-$3_{↓}1_{↑}2_{↓}$+N-term α; Right) = 16.0, and OF(Extended-$3_{↓}1_{↑}2_{↓}$+N-term α; Left) = 2.5, leading to the ratios,
(3)$$\begin{array}{ccc} \frac{OF(\mathrm{Extended}-1_{↑}3_{↓}2_{↑}+C-\mathrm{term}\;\alpha ;\mathrm{Left})}{OF(\mathrm{Extended}-1_{↑}3_{↓}2_{↑}+C-\mathrm{term}\;\alpha ;\mathrm{Right})} & \approx & 0.0068,\\ \frac{OF(\mathrm{Extended}-3_{↓}1_{↑}2_{↓}+N-\mathrm{term}\;\alpha ;\mathrm{Left})}{OF(\mathrm{Extended}-3_{↓}1_{↑}2_{↓}+N-\mathrm{term}\;\alpha ;\mathrm{Right})} & \approx & 0.156, \end{array}$$
suggesting that some mechanism exists for enhancing the occurrence of the left-handed βαβ-unit in the $3_{↓}1_{↑}2_{↓}+N-\mathrm{term}\;\alpha$ structure. A plausible explanation is that the left-handed βαβ-unit was chosen in these domains to avoid the collision between two helices lying on the same side of the β-sheet in the $\mathrm{Extended}-3_{↓}1_{↑}2_{↓}+N-\mathrm{term}\;\alpha$ structures. This mechanism suggests that the rule for minimizing frustration between the structural preferences of secondary structures lying nearby on the same side of the β-sheet is comparably strong as the rule of the right-handedness of the βαβ-unit.

### 3.3. Frustration and Function

A remaining critical question is the reason for the existence of protein domains having the reverse ferredoxin topology. Because proteins have evolved not for their stability but their functions, a possible explanation is that frustrated structures are necessary for their functions. Roles of frustration in functions have been discussed with theoretical methods by inferring the local degree of frustration using the coarse-grained energy function of protein conformation [39]. By computationally perturbing the sequence or configuration of a local part of the protein, the local part was regarded as less frustrated when most of the perturbations increase the calculated free energy significantly, while the local part was regarded as frustrated when the free energy change upon perturbations is insignificant [40]. It was shown that the local frustration can guide thermal motions [41] and specific associations [42], suggesting the positive role of frustration in protein functioning.

In this study, we proposed a new definition of frustration as the conflict between structural preferences of local parts of the protein. This definition of frustration should shed further light on the role of frustration. The frustrating interaction between helices in the reverse ferredoxin fold destabilizes the structure. This tendency may be compensated for by a specific residue design to stabilize the fold, or the protein may utilize the tendency to enhance the fluctuation and facilitate the structural change, which is needed for its functioning. An example shown in Figure 1B was the catalytic core of human DNA polymerase kappa. Because the sizeable structural change is necessary for activating a molecular motor motion of DNA polymerase, we can expect that the frustration in this structure helps function DNA polymerase.

The definition of frustration introduced in this study, the structural conflict among the local parts’ structural preferences, provides a new perspective to the frustration-function relationship. In particular, the hypothesis proposed in this subsection suggests an intriguing possibility that the designed incorporation of frustration in the structure helps design the protein whose function is related to mobility with the significant structural change. To test this hypothesis, the dynamics and stability of the frustrated proteins and the specific design of sequences to fold the frustrated structures should be examined with further direct and systematic methods.

## 4. Materials and Methods

### 4.1. Detecting the Position of the C/N Terminal α-Helix

We explain in Figure 6 the method to judge on which side of the β-sheet plane the C or N-terminal α-helix lies in protein domains. We defined three vectors, **a**, **b**, and **c** in the $1_{↑}3_{↓}2_{↑}+C-\mathrm{term}\;\alpha$ (Figure 6A) and $3_{↓}1_{↑}2_{↓}+N-\mathrm{term}\;\alpha$ (Figure 6B) structures. The terminal α-helix is on the upper side of the β-sheet plane of Figure 6 if (a×b)·c>0 and the helix is on the lower side of the plane if (a×b)·c<0.

### 4.2. Definition of the Distance *x* between the Plane of β-Pleats and the α-Helix in the αβ- or βα-Unit

We measured the distance *x* between the plane of β-pleats and the α-helix in the αβ- and βα-units by introducing a xyz-coordinate system in each unit (Figure 7). For defining the coordinate system, we set the direction of the *y*-axis parallel to the β-strand axis, and set the *y*-*z* plane parallel to the plane defined by the terminal three Cα atoms of the β-strand. We set the direction of the *z*-axis so as to place the helix on the z>0 side. This idea of the coordinate system can be described in a precise way by defining the basis vectors, $\vec{e_{x}}$, $\vec{e_{y}}$, and $\vec{e_{z}}$, of the xyz-coordinate system with $\vec{e_{z}}$ being $\vec{e_{z}}=\vec{e_{x}}\times \vec{e_{y}}$.

We defined $\vec{e_{x}}$ and $\vec{e_{y}}$ as in the following way. Let *i* be the number of the terminal residue of the β-strand (the C-terminal residue in the βα-unit and the N-terminal residue in the αβ-unit) and $C\alpha_{i}$ be the position of the *i*th Cα atom. We defined $\vec{e_{x}}$ by categorizing the βα- or αβ-unit into two types, the parallel and antiparallel unit (Figure 7A,B). Then, we defined $\vec{e_{x}}$ as a normalized vector having the direction, which places both the starting and ending points of the α-helix on the coordinate of x>0;
(4)$$\begin{array}{c} \vec{e_{x}}\|\left\{\begin{array}{cc} \vec{C\alpha_{i-2}C\alpha_{i-1}}\times \vec{C\alpha_{i-1}C\alpha_{i}} & (\mathrm{parallel}\;\beta \alpha -\mathrm{unit}),\\ \vec{C\alpha_{i}C\alpha_{i-1}}\times \vec{C\alpha_{i-1}C\alpha_{i-2}} & (\mathrm{antiparallel}\;\beta \alpha -\mathrm{unit}),\\ \vec{C\alpha_{i}C\alpha_{i+1}}\times \vec{C\alpha_{i+1}C\alpha_{i+2}} & (\mathrm{parallel}\;\alpha \beta -\mathrm{unit}),\\ \vec{C\alpha_{i+2}C\alpha_{i+1}}\times \vec{C\alpha_{i+1}C\alpha_{i}} & (\mathrm{antiparallel}\;\alpha \beta -\mathrm{unit}), \end{array}\right. \end{array}$$
and $\vec{e_{y}}$ is a normalized vector, whose direction is
(5)$$\begin{array}{c} \vec{e_{y}}\|\left\{\begin{array}{cc} \vec{C\alpha_{i-2}C\alpha_{i}} & (\beta \alpha -\mathrm{unit}),\\ \vec{C\alpha_{i+2}C\alpha_{i}} & (\alpha \beta -\mathrm{unit}). \end{array}\right. \end{array}$$

### 4.3. Rosetta Folding Simulations

We performed the Rosetta folding simulations to test the realizability of the blueprint structures. Here, Rosetta is a software suite that includes algorithms for macromolecular modeling, docking, protein design, etc [43]. Among the many algorithms included in the Rosetta software, we used the Rosetta BluePrintBDR protocol [43] for folding simulations. With this protocol, we performed the folding simulations by assembling one, three, or nine-residue length fragments so as to make the assembled structure compatible with a “blueprint”, which describes the length of the secondary structure elements, strand pairings, and backbone torsion ranges for each residue. In thsese simulations, the main chain was represented by N, NH, Cα, C, and CO, and the side chain was represented by a sphere using the centroid model of Rosetta. We used the simulated annealing method to search for low-energy structures, and recorded the last structure of each simulated annealing run as a compatible structure only when the structure met the conditions specified in the blueprint.

As in models of Ref. [44], we represented all the residues as Valine, and used the same energy parameters as in Ref. [44]. The use of the poly-Valine sequence is because our purpose is to determine whether the phenomena observed in the database are explained by backbone properties rather than by the sequence-specific properties. Valine is the smallest and strongest hydrophobic amino acid, which suits this purpose, as shown in Ref. [5]. Figure 8 shows the blueprints we used in the BluePrintBDR protocol. In these blueprints, we used the same length of secondary structures and loops as optimized in Ref. [44]. The purpose of the present Rosetta simulations is to analyze the statistical tendency among different topologies. Because loops in each topology are shorter than five-residue length in most folds, and their distribution is peaked at around the two- to three-residue length (Figure S5), it is sufficient to use the short loops in the blueprints. Here, for the computational efficiency, we restricted ourselves to the loops with two- to three residue length for βα- and αβ-loops. For β-hairpin loops, we assumed that loop consists of two, four, or five residues in the blueprints because the two or five-residue length is necessary for keeping the chirality rule of the hairpin loop [5] (Figure 8).

In the folding simulations, we did not impose the ABEGO constraint on the loop regions, but we imposed the constraint on the secondary structure regions by making the dihedral angles of the main chain in these regions fall into the ABEGO classes compatible with the secondary structures designated by the bluprint. Here, the ABEGO classification is a coarse-grained representation of the dihedral angles, specifying the regions in a Ramachandran plot with the alphabetic symbols: A, B, E, G, and O denote the right-handed α-helix region, right-handed β-strand region, left-handed β-strand region, left-handed helix region, and the cis peptide conformation, respectively [45].

### 4.4. Score to Detect the Left-Handed βαβ-Unit

We detected protein domains having the left-handed βαβ-unit by calculating the score of the left-handedness (*L*-score). Here, for defining the *L*-score, we consider a βαβ-unit exemplified in Figure 9A. We refer to the N-terminal β-strand in the βαβ-unit as β1, and the C-terminal β-strand as β2. We should note that the following *L*-score is applicable to evaluating the left-handedness of structures in which β1 and β2 are not connected directly to each other by hydrogen bonds, but multiple β-strands intervene between β1 and β2. We write the residue length of β1, β2, and the linker part connecting β1 and β2 as *n*, *m*, and *l*, respectively. We label the residues in those parts as $\left(N_{1},N_{2},\cdots ,N_{n}\right)$, $\left(C_{1},C_{2},\cdots ,C_{m}\right)$, and $\left(L_{1},L_{2},\cdots ,L_{l}\right)$.

We define the residue number $C_{\max}\left(N_{i},N_{i+1}\right)$ so as to maximize the peak angle in Figure 9B when the residues $N_{i}$ and $N_{i+1}$ are given. Similarly, we define the residue number $N_{\max}\left(C_{j},C_{j+1}\right)$ to maximize the peak angle;
(6)$$\begin{array}{ccc} C_{\max}\left(N_{i},N_{i+1}\right) & = & \arg\max_{C_{j}}\left[∠C\alpha_{N_{i}}C\alpha_{C_{j}}C\alpha_{N_{i+1}}\right],\\ N_{\max}\left(C_{j},C_{j+1}\right) & = & \arg\max_{N_{i}}\left[∠C\alpha_{C_{j}}C\alpha_{N_{i}}C\alpha_{C_{j+1}}\right]. \end{array}$$

Then, using the Heaviside function, H[x]=1 for x>0 and H[x]=0 for x≤0, the *L*-score is defined as
(7)$$\begin{array}{ccc} L-score & = & \frac{1}{[(n-1)+(m-1)]\cdot l}\sum_{k=1}^{l}\left[\sum_{i=1}^{n-1}H\left[\left(\vec{C\alpha_{N_{i}}C\alpha_{N_{i+1}}}\times \vec{C\alpha_{N_{i}}C\alpha_{C_{\max}\left(N_{i},N_{i+1}\right)}}\right)\cdot \vec{C\alpha_{N_{i}}C\alpha_{L_{k}}}\right]\right.\\ & + & \left.\sum_{j=1}^{m-1}H\left[\left(\vec{C\alpha_{C_{j+1}}C\alpha_{C_{j}}}\times \vec{C\alpha_{C_{j}}C\alpha_{N_{\max}\left(C_{j},C_{j+1}\right)}}\right)\cdot \vec{C\alpha_{C_{j}}C\alpha_{L_{k}}}\right]\right]. \end{array}$$

The *L*-score ranges from 0 to 1 (Figure 9C). The higher the score, the more left-handed the βαβ-unit becomes. We judged the unit is left-handed when *L*-score≥0.6.

## Figures and Tables

**Figure 1 molecules-27-03547-f001:**
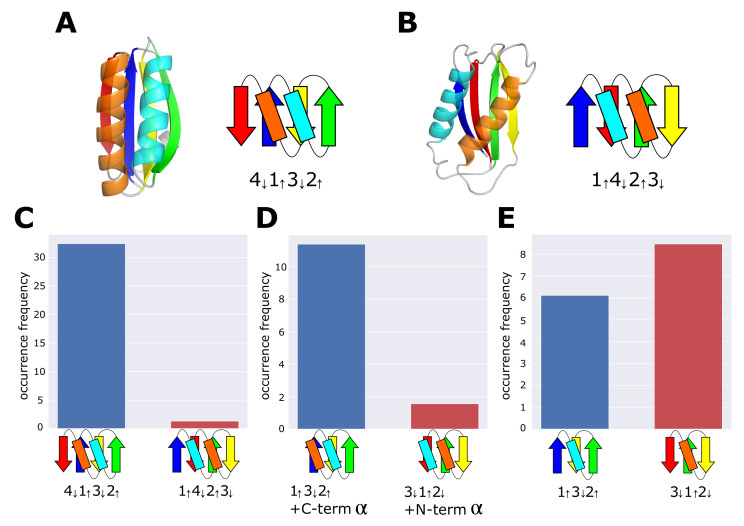
Topology and occurrence frequency of the ferredoxin fold and the reverse ferredoxin fold. (**A**) An example structure (a microcompartment protein, PDB ID: 4QIV) and the topology $4_{↓}1_{↑}3_{↓}2_{↑}$ of the ferredoxin fold. (**B**) An example structure (the catalytic core of human DNA polymerase kappa, PDB ID: 1T94) and the topology $1_{↑}4_{↓}2_{↑}3_{↓}$ of the reverse ferredoxin fold. (**C**) Occurrence frequency of the ferredoxin topology $4_{↓}1_{↑}3_{↓}2_{↑}$ and the reverse ferredoxin topology $1_{↑}4_{↓}2_{↑}3_{↓}$. (**D**) Occurrence frequency of the topology $1_{↑}3_{↓}2_{↑}+C-\mathrm{term}\;\alpha$ and the topology $3_{↓}1_{↑}2_{↓}+N-\mathrm{term}\;\alpha$. (**E**) Occurrence frequency of the topology $1_{↑}3_{↓}2_{↑}$ and the topology $3_{↓}1_{↑}2_{↓}$. In (**C**–**E**), the dataset of the 99% sequence identity representatives derived from ECOD was used. Chains are colored from blue (N-terminus) to red (C-terminus). In the topology diagram, β-strands are represented with arrows and α-helices are rectangles.

**Figure 2 molecules-27-03547-f002:**
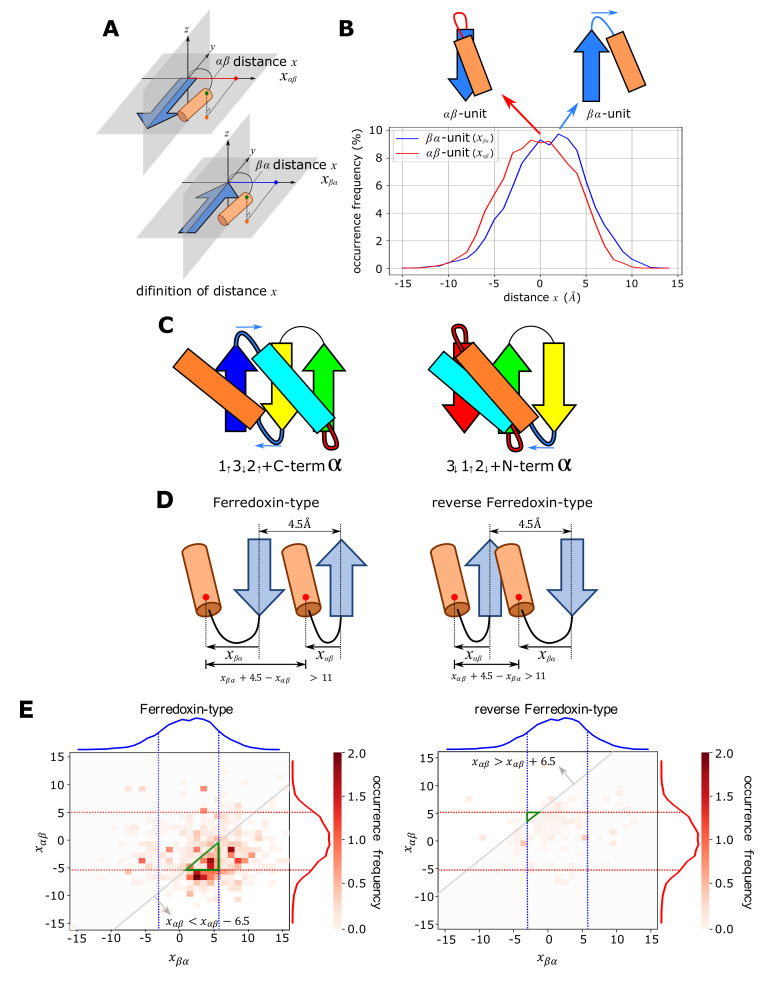
Absence or presence of the structural conflict between α-helices. (**A**) Definition of the distance *x* between the pleated plane of the β-strand and the α-helix in the αβ-unit (top) and the βα-unit (bottom). (**B**) Distribution of *x* in the αβ-unit (red) and the βα-unit (blue). The distribution was found in the culled PDB dataset with the parameters of the sequence identity <30%, the finer resolution than 2.0 Å, and the *R*-factor <0.25. (**C**) Structural preferences of the the αβ-unit (connected by a red linker) and the βα-unit (connected by a blue linker) prevent collision between the terminal helix and the helix in the βαβ structure in the $1_{↑}3_{↓}2_{↑}+C-\mathrm{term}\;\alpha$ topology (left), while they induce a collision in the $3_{↓}1_{↑}2_{↓}+N-\mathrm{term}\;\alpha$ topology (right). Blue arrows show the shift of α-helix induced by the x>0 preference of the βα-unit. (**D**) The necessary condition to avoid the collision of two helices. $x_{\beta \alpha }-x_{\alpha \beta }+4.5\;Å>11$Å for the ferredoxin-type topology and $x_{\alpha \beta }-x_{\beta \alpha }+4.5\;Å>11$Å for the reverse ferredoxin-type topology. (**E**) The realizable area to avoid the collision and the occurrence frequency of $\left(x_{\beta \alpha },x_{\alpha \beta }\right)$ in the ECOD database. The realizable area satisfying the three conditions; the necessary condition to avoid the collision, the condition of the frequency >5% in the $x_{\beta \alpha }$ distribution, and the condition of the frequency >5% in the $x_{\alpha \beta }$ distribution; is shown with a green triangle on the $\left(x_{\beta \alpha },x_{\alpha \beta }\right)$ plane. The occurrence frequency shown with the gray-scale is superposed. Blue and red curves are distributions in (**B**).

**Figure 3 molecules-27-03547-f003:**
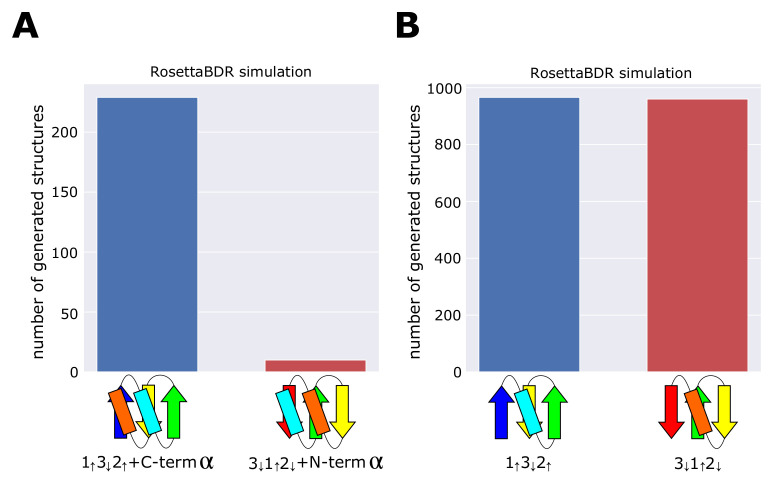
The number of simulated structures compatible with the blueprint. We repeated the Rosetta folding simulations 10,000 times and counted the number of compatible structures generated. (**A**) Comparison between the $1_{↑}3_{↓}2_{↑}+C-\mathrm{term}\;\alpha$ topology and the $3_{↓}1_{↑}2_{↓}+N-\mathrm{term}\;\alpha$ topology. In simulations, the number of structures in which two helices lie on the same side of the β-sheet surface was counted. (**B**) Comparison between the $1_{↑}3_{↓}2_{↑}$ topology and the $3_{↓}1_{↑}2_{↓}$ topology.

**Figure 4 molecules-27-03547-f004:**
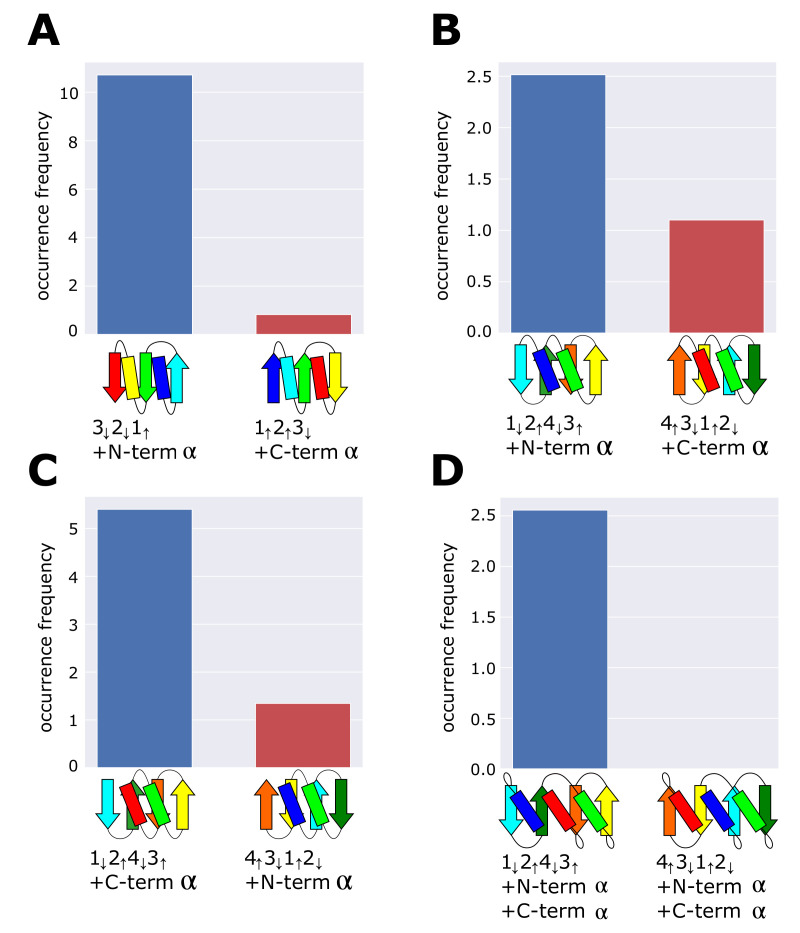
Comparisons of occurrence frequency between topologies minimizing frustration and their reverse topologies exhibiting frustration. (**A**) $3_{↓}2_{↓}1_{↑}+N-\mathrm{term}\;\alpha$ and $1_{↑}2_{↑}3_{↓}+C-\mathrm{term}\;\alpha$, (**B**) $1_{↓}2_{↑}4_{↓}3_{↑}+N-\mathrm{term}\;\alpha$ and $4_{↑}3_{↓}1_{↑}2_{↓}+C-\mathrm{term}\;\alpha$, (**C**) $1_{↓}2_{↑}4_{↓}3_{↑}+C-\mathrm{term}\;\alpha$ and $4_{↑}3_{↓}1_{↑}2_{↓}+N-\mathrm{term}\;\alpha$, and (**D**) $1_{↓}2_{↑}4_{↓}3_{↑}+N-\mathrm{term}\;\alpha +C-\mathrm{term}\;\alpha$ and $4_{↑}3_{↓}1_{↑}2_{↓}+N-\mathrm{term}\;\alpha +C-\mathrm{term}\;\alpha$. The dataset was the 99% sequence identity representatives derived from the ECOD database.

**Figure 5 molecules-27-03547-f005:**
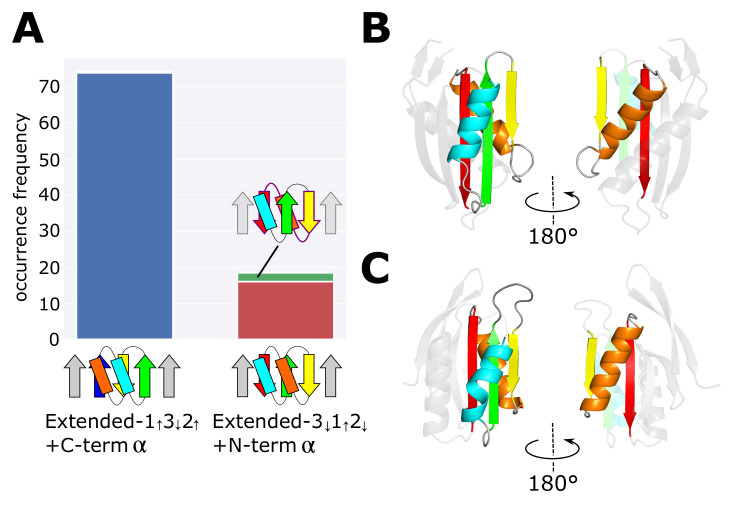
Occurrence of the left-handed and right-handed βαβ-units in the extended domains which include the $1_{↑}3_{↓}2_{↑}+C-\mathrm{term}\;\alpha$ or $3_{↓}1_{↑}2_{↓}+N-\mathrm{term}\;\alpha$ structure. (**A**) Comparison between occurrence frequencies of extended domains that include the $1_{↑}3_{↓}2_{↑}+C-\mathrm{term}\;\alpha$ or $3_{↓}1_{↑}2_{↓}+N-\mathrm{term}\;\alpha$ as the partial structure. The occurrence frequency of the extended $1_{↑}3_{↓}2_{↑}+C-\mathrm{term}\;\alpha$ is 74.3 among which the occurrence frequency of structures having the left-handed βαβ-unit is 0.5 (invisible in the figure). The occurrence frequency of the extended $3_{↓}1_{↑}2_{↓}+N-\mathrm{term}\;\alpha$ structure is 18.5 among which the occurrence frequency of structures having the left-handed βαβ-unit is 2.5 (green). (**B**,**C**) Examples of the extended $3_{↓}1_{↑}2_{↓}+N-\mathrm{term}\;\alpha$ domains having the left-handed βαβ-unit. (**B**) PDB ID: 2CVE. (**C**) PDB ID: 1RLH.

**Figure 6 molecules-27-03547-f006:**
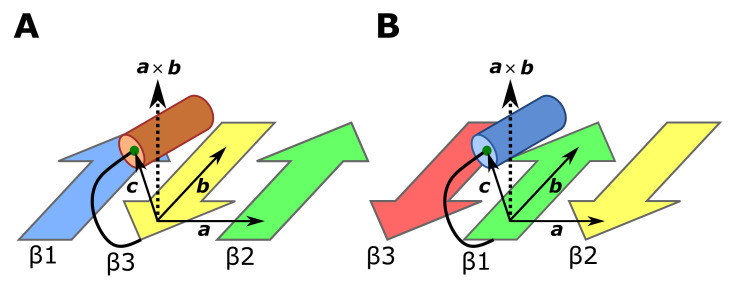
The method to judge on which side of the β-sheet the C or N-terminal α-helix lies. We defined three vectors, **a**, **b**, and **c**. The helix lies on the upper side of the β-sheet plane if (a×b)·c>0 and the helix lies on the lower side of the plane if (a×b)·c<0. (**A**) In the $1_{↑}3_{↓}2_{↑}+C-\mathrm{term}\;\alpha$ structure, the vector **a** is a vector extending from the Cα atom of the C-terminal residue of the β-strand 3 (yellow arrow) to the Cα atom of the N-terminal residue of the β-strand 2 (green arrow). The vector **b** is a vector extending from the Cα atom of the C-terminal residue of the β-strand 3 to the Cα atom of the second residue before the C-terminal residue of the β-strand 3. The vector **c** is a vector extending from the Cα atom of the C-terminal residue of the β-strand 3 to the center of mass (green dot) of Cα atoms of four N-terminal residues of the α-helix (orange cylinder). (**B**) In the $3_{↓}1_{↑}2_{↓}+N-\mathrm{term}\;\alpha$ structure, the vector **a** is a vector extending from the Cα atom of the N-terminal residue of the β-strand 1 (green arrow) to the Cα atom of the C-terminal residue of the β-strand 2 (yellow arrow). The vector **b** is a vector extending from the Cα atom of the N-terminal residue of the β-strand 1 to the Cα atom of the second residue after the N-terminal residue of the β-strand 1. The vector **c** is a vector extending from the Cα atom of the N-terminal residue of the β-strand 1 to the center of mass (green dot) of Cα atoms of four C-terminal residues of the α-helix (blue cylinder).

**Figure 7 molecules-27-03547-f007:**
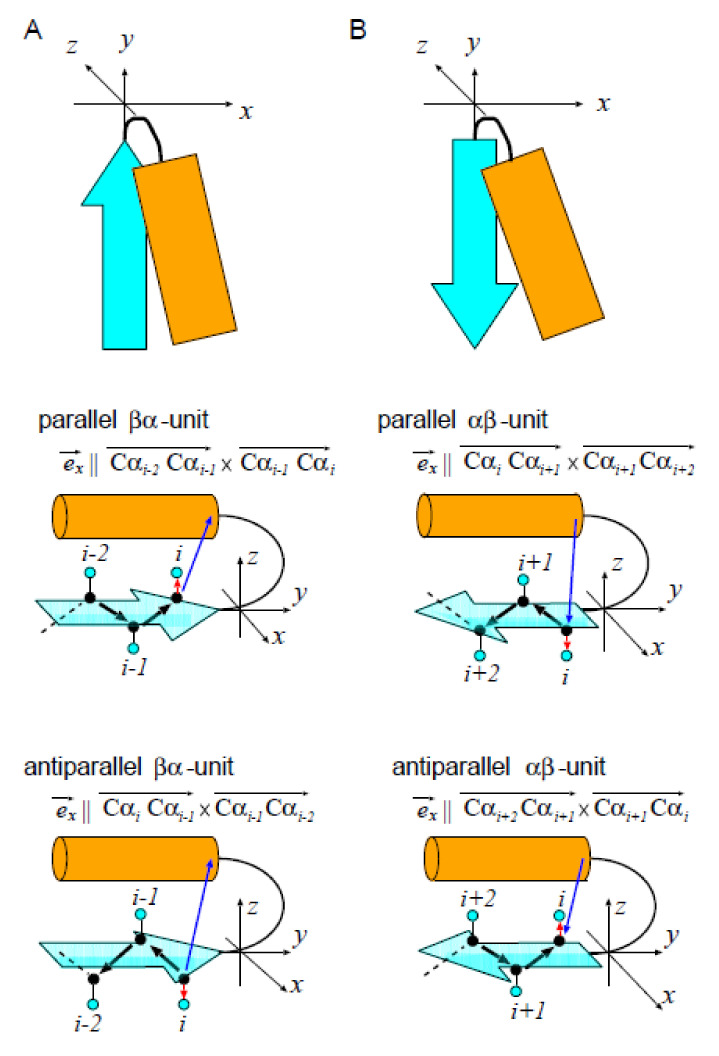
The xyz-coordinate system to define the distance *x* between the plane of β-pleats and the α-helix. (**A**) The βα-unit and (**B**) the αβ-unit. These units consist of a β-strand (cyan arrow) and an α-helix (orange rectangle). Top panels represent the rough sketch of the coordinate system. Middle and bottom panels show Cα atoms (black dots), Cβ atoms (cyan dots), a vector spanning from the Cα to the Cβ of the terminal residue of the β-strand (i.e., the residue in the strand nearest to the helix) in each unit (red arrow), and a vector spanning from the Cα of the terminal residue of the β-strand to the center of mass of terminal four residues of the α-helix (i.e., four residues in the helix nearest to the strand) in each unit. Unit is referred to as “parallel” when the inner product of red and blue arrows is positive, and as “antiparallel” when the inner product is negative.

**Figure 8 molecules-27-03547-f008:**
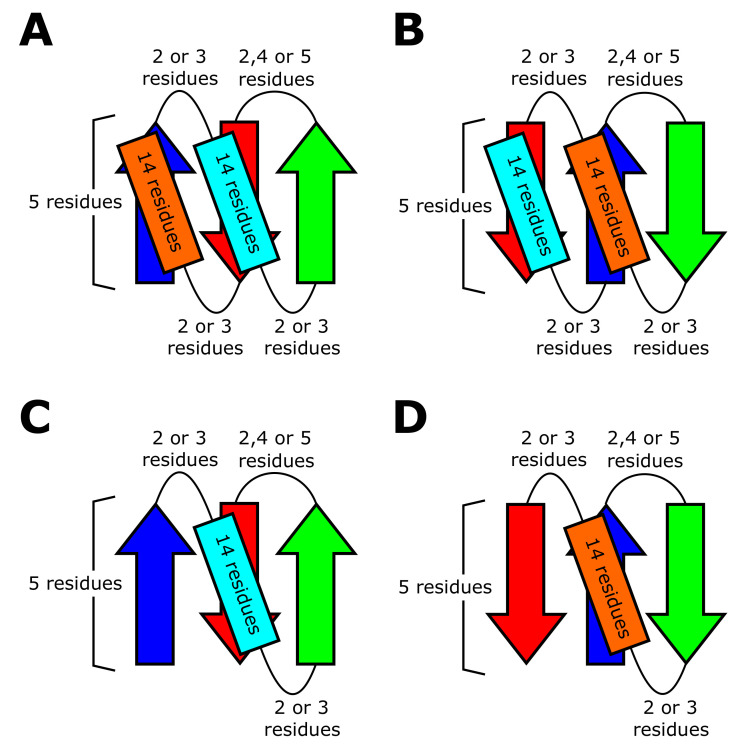
Blueprints used in the Rosetta folding simulations. The blueprints are represented by β-strands (arrows), α-helices (rectangles), and loops (curved lines). Blueprints of (**A**) the $1_{↑}3_{↓}2_{↑}+C-\mathrm{term}\;\alpha$ topology, (**B**) the $3_{↓}1_{↑}2_{↓}+N-\mathrm{term}\;\alpha$ topology, (**C**) the $1_{↑}3_{↓}2_{↑}$ topology, and (**D**) the $3_{↓}1_{↑}2_{↓}$ topology.

**Figure 9 molecules-27-03547-f009:**
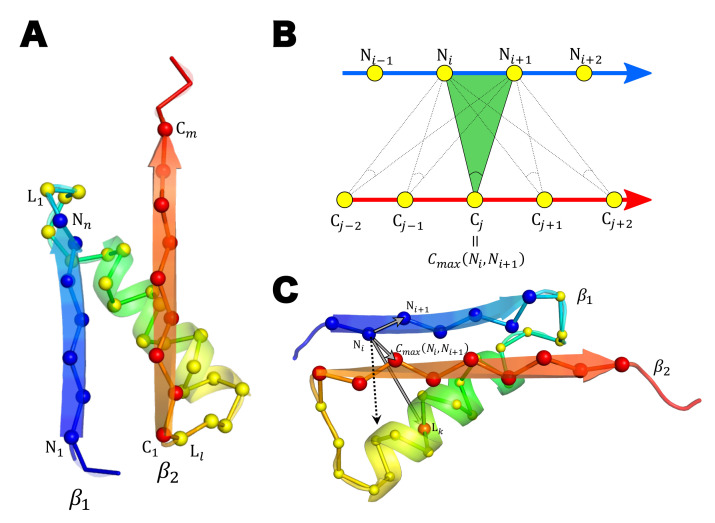
Calculation of the left-handedness score, L-score. (**A**) An example left-handed βαβ-unit. The cartoon representation and the backbone representation of the main chain are superposed. Cα atoms are drawn with spheres in the backbone representation. The first and the last residue numbers of β1, β2, and the linker part are labeled on the chain. (**B**) Determination of $C_{\max}\left(N_{i},N_{i+1}\right)$. (**C**) Calculation of a term in L-score. The vector connecting $C\alpha_{N_{i}}C\alpha_{N_{i+1}}$, the one connecting $C\alpha_{N_{i}}C\alpha_{C_{\max}\left(N_{i},N_{i+1}\right)}$, and the one connecting $C\alpha_{N_{i}}C\alpha_{L_{k}}$ in Equation (Equation 7) are drawn with gray arrows and the vector product of the first two vectors are drawn with a dashed arrow. The calculated score of this example βαβ-unit is L-score=0.86.

## Data Availability

Not applicable.

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
