# Peer review of "The Structural Rule Distinguishing a Superfold: A Case Study of Ferredoxin Fold and the Reverse Ferredoxin Fold"

_molecules, 2022, doi:10.3390/molecules27113547_

Round 1
Reviewer 1 Report
Although the goal of explaining the folding and superfolding of proteins isa topic that deserves attention due to the implications related to the
correct folding that proteins assume also in function of implications
related to pathologies, however the manuscript remains a speculative work
exclusively of bioinformatics which requires an in-depth study with more
direct studies on the proteins taken into consideration.
For this my opinion is to reject.
Reviewer 2 Report
The manuscript by Nishina and coworkers aims to distinguish superfolds from the other folds, using as model the typical ferredoxin fold vs the so call reverse ferredoxin fold. The authors base their conclusions on the number of occurring structures in known databases. Then they use Rosetta simulations, to assess structural conflicts, but they have substituted all residues in the model to valine, a restriction that can influence the result obtained. Moreover, the relation between frustration and function that the authors discuss in the end of the manuscript, given only one example, is not new. Overall the paper does not make important advances on the field.
Reviewer 3 Report
The manuscript discusses the cause of the prevalence of the Ferredoxin fold compared to its reverse fold. It is an interesting study. Fig. 2 is striking, and the results of the Rosetta simulation reinforce the conclusion.
However, I cannot stop wondering how I can understand the results. I do not find any phrase trying to explain the physical basis of the difference of the two folds presented in Fig. 2-4, or what kind of interactions make the difference. Without any clue to the cause, it is simply a mystery. Usually, simulation is used to resolve the mechanism behind complex observation. In this study, however, the Rosetta simulation was treated as another experiment as if it did not allow to go into details.
This impression first came from Fig. 2. The distributions of the alpha/beta-unit and the beta/alpha-unit are beautifully shifted by 2-3A throughout the whole range of the distance, though the range of the distance extend more than 20A. There are two points here: (1) The difference in the distance distributions seems to be not large enough to decisively distinguish the authentic fold from the reverse fold, compared to the clear distinction in the Rosetta simulation (Fig. 3); (2) The crystal structures of the folds, in addition to the two in Fig.1, have a large variety in the length of the loop connecting the alpha helix to the beta strand, and also a large variety in the orientation between the alpha helix to the beta strand. The beautiful shift in Fig. 2 and the seemingly less consistent results together with the full variety of the structures puzzled me. I hope that the authors provide some reasonable explanation to this problem.
The followings are each specific point.
(1) Fig. 1C, D, and E
What dataset in the ECOD was used to obtain these frequencies? The authentic fold may correspond to “X: Alpha-beta plaits”, the reverse fold to “X: Reverse ferredoxin”. Are these all for Fig. 1C? What are the datasets for Fig. 1D and E, which may not form a superfold?
“X: Reverse ferredoxin” in the ECOD contains four groups of “Homology level.” Is this consistent with the statement in the text, “the reverse ferredoxin fold is found only in one superfamily”?
(2) Fig. 2B
What dataset was used to obtain these frequencies. Were these obtained from the same dataset as those used in Fig. 1, or using a larger set?
(3) Fig. 2C
In the right cartoon, the C-term helix shifts parallel to the N-term helix. However, the other end of the C-term helix having the ab-unit will keep the distance. Thus, a kind of rotation is more suitable drawing for the shift of the C-term helix. It is trivial, but better have a proper description.
(4) Fig. 4
The same question as in (1) and (2). What dataset in the ECOD was used to obtain these frequencies?
(5) Scheme 3 (above Fig. 5)
What are the real numbers of the structures to calculate the ratios, 0.0068 and 0.156?
(6) 3.3. Frustration and function
The stability or dynamics of the reverse fold is out of the scope of the manuscript. If the authors wish to keep this statement, it is necessary to add certain data that support it.
As far as I understand, the superfold is the structure that has a sizable capacity of allowing the sequence variation. It may be possible to say that the reverse fold has a highly specific sequence to maintain a sufficient level of stability.
(7) 4.3. Rosetta folding simulations
General readers, not in the protein modeling field, do not know what is done by Rosetta. A brief introduction to Rosetta is helpful for a better understanding. Further, BluePrintBDR protocol and ABEGO also need short explanations.
(8) Fig. 8
I wonder if the short length of the loop between alpha and beta resulted in the clear separation between the two fold. What is the reason why the loop length is limited to 2-3?
Reviewer 4 Report
This manuscript outlines a plausible explanation for the difference between superfolds and regular folds by analyzing the preferred positioning of helices in the ferredoxin superfold and the reverse ferredoxin fold, which is a regular fold. Through careful analysis of data in the PDB, the authors find that the preferred position of the N-term helix of the reverse ferredoxin fold leads to a steric clash with the other helix in the fold. An analogous clash does not occur in the ferredoxin superfold. The authors propose a new criteria to distinguish superfolds from regular folds: minimal structural frustration or conflict among secondary structures. As a further test of this hypothesis, they show that many fewer compatible structures are produced by the Rosetta design algorithm when the helix clash is possible. This study should be of strong appeal to those interested in protein design and the evolution of folds. A few comments follow:
- Near the end of section 2.1 (lines 124 -129), where the occurrence of helices on the same versus the opposite side of the beta sheet is discussed, it might be more useful to report the ratio of the integrated areas of the observed frequencies of the helices being on the opposite (theta < 0) versus the same (theta > 0) side of the 1(up)3(down)2(up) + C-term a and 3(down)1(up)2(down) + N-term a for the data in Figure S2C. This would point out that the preference for the helices to be on the opposite side of the beta sheet is much higher for the 3(down)1(up)2(down) + N-term a topology.
- If possible, it would be useful to see what the plot in Figure 2C looks like for data used to generate Figure S2C for the 3(down)1(up)2(down) + N-term a fold with the helices on the same versus the opposite side of the beta sheet. When the helices are on the same side of the sheet, the steric conflict must be resolved so the distribution should look more like the 1(up)3(down)2(up) + C-term a. If this is the case, then the question is at what cost? Is the packing against the beta sheet poorer for the N-term helix when the steric clash is resolved by moving the helix out of its preferred position?
- In section 2.3 line 168, it states that 10,000 Rosetta simulations were done. In the caption to Figure 3 it states 1000. Which is correct?
- In Figures 6 and S1, it would be helpful for readers to label the beta sheets (as they are in Figure S2).
Round 2
Reviewer 1 Report
The authors revised the manuscript and made a considerable improvement. In particular, explanations and simplifying examples have been introduced in the results section and in the methods to fully understand the methodology and proposed results. The figures and additional explanations in the captions also help to make the meaning of the results clearer. Therefore I believe that the work can be considered acceptable in this form.
Reviewer 2 Report
The authors revised the manuscript accordingly to suggestions, improving its content.